# Insecticide resistance and species diversity in *Anopheles gambiae* s.l. in Côte d'Ivoire

Rosine Z. Wolie[1,2,3]*, Alphonsine A. Koffi[2,3], Joseph Chabi[2], Ludovic P. Ahoua Alou[2,3], Eleanore D. Sternberg[4,5], Kouakou P. M. Kouame[2], Florent H. A. Yapo[2], Soromane Camara[2,3], Amal Dahounto[2], Innocent Z. Tia[2,3,6], Welbeck A. Oumbouke[2,7], Kallista Chan[8], Oulo N'Nan-Alla[9], Raphael N'Guessan[2,3,8]

**1** Université Nangui Abrogoua, UFR Des Sciences de la Nature, Abidjan, Côte d'Ivoire, **2** Vector Control Product Evaluation Centre, Institut Pierre Richet (VCPEC-IPR), Bouaké, Côte d'Ivoire, **3** Institut Pierre Richet (IPR)/Institut National de santé Publique (INSP), Bouaké, Côte d'Ivoire, **4** Tropical Health LLP, London, United Kingdom, **5** Department of Vector Biology, Liverpool School of Tropical Medicine, Liverpool, United Kingdom, **6** Université Alassane Ouattara, Bouaké, Côte d'Ivoire, **7** Vector Biology Department, Liverpool School of Tropical Medicine, Liverpool, United Kingdom, **8** Department of Disease Control, London School of Hygiene and Tropical Medicine, London, United Kingdom, **9** Université Félix Houphouët-Boigny, UFR Biosciences, Unité de Recherche et de Pédagogie de Génétique, Abidjan, Côte d'Ivoire

* wolierosine@yahoo.fr

## Abstract

### Background

Insecticide resistance in malaria vectors poses a significant threat to effective control of malaria vectors across sub-Saharan Africa. In Côte d'Ivoire, *Anopheles gambiae* s.l. exhibits considerable variability in species distribution and insecticide resistance levels across distinct ecological areas. This study reports on insecticide resistance profiles within *An. gambiae* complex as well as species composition dynamics in various ecological settings in Côte d'Ivoire.

### Methods

From July to October 2020, larval and adult mosquitoes were collected across three ecological settings (savannah, pre-forest and forest) in Côte d'Ivoire. Larvae were reared to adult stage for susceptibility bioassays. Diagnostic concentrations (DCs) applied against *An. gambiae* s.l. were deltamethrin 0.05%, permethrin 0.75% and pirimiphos methyl 0.25% using WHO susceptibility test kits. When increased survival to diagnostic concentrations (DCs) was observed, intensity bioassays were conducted using 5x and 10x DCs. In addition, synergist assays were performed at the DC level with pre-exposure to 4% piperonyl butoxide (PBO). Together with adults directly collected from the field, female samples from the bioassays outcome were speciated and resistance target site mechanisms (*Kdr* L1014F and *Ace-1*R G119S)

**Data availability statement:** All relevant data are available within the manuscript and its Supporting Information files. The underlying datasets generated during the study have been deposited in Zenodo and are publicly accessible at: https://doi.org/10.5281/zenodo.16953434.

**Funding:** This work was supported by a grant awarded to the Institut Pierre Richet / Institut National de Santé Publique, Côte d'Ivoire, from the Bill & Melinda Gates Foundation (OPP1210335) through the Pan African Mosquito Control Association (PAMCA). The grant aimed to strengthen malaria vector surveillance in Côte d'Ivoire, specifically addressing the genetic diversity and gene flow data gaps in insecticide-resistant Anopheles gambiae. The funders had no role in study design, data collection and analysis, decision to publish, or preparation of the manuscript.

**Competing interests:** The authors have declared that no competing interests exist.

**Abbreviations:** WHO, World Health Organization; LLINs, long lasting insecticidal nets; $L1014F$ Kdr, west knockdown resistance; Ace-1R, acetylcholinesterase-1 resistance; VCPEC, Vector Control Product Evaluation Centre; IPR, Institut Pierre Richet; Ace-1$^R$ $G119S$, G119S mutation in Ace-1$^R$; R, resistant; S, susceptible.

were determined using PCR. Species distribution and insecticide resistance were analyzed across ecological areas.

## Results

Significant variations in insecticide resistance phenotypes and mechanisms in *An. gambiae* s.l. were found across the three ecological areas in Côte d'Ivoire. Mortality rates, following pyrethroid exposure, were significantly higher in *An. gambiae* s.l. collected in the savannah zone compared to those collected from the forest zone ($p < 0.05$). *An. gambiae* was more predominant in the savannah area (98.8%, 95% confidence of interval (CI) [93.5–100.0]), whereas *An. coluzzii* dominated in the pre-forest (92.9%, 95% CI [88.9–95.8]) and forest zones (61.3%, 95% CI [53.4–68.9]). Such predominance of *An. coluzzii* was associated with high intensity of pyrethroid resistance in these areas. Overall, the allelic frequencies of the resistance mutations in *An. gambiae* were higher than those in *An. coluzzii* regardless of the area. *Kdr* L1014F frequency in the forest zone was 78.1% (95% CI [68.0–88.2]) in *An. gambiae* larvae, 51.8% (95% CI [33.3–70.4]) in adults, and 63.1% (95% CI [52.4–73.8]) in *An. coluzzii* larvae.

## Conclusion

Variations in species distribution and insecticide resistance in ecological areas in Côte d'Ivoire should be carefully considered when developing and implementing vector control strategies.

## Background

Malaria is a mosquito-borne disease that affects hundreds of millions of people annually and remains a leading cause of mortality in the developing world, imposing significant human suffering and substantial socioeconomic burdens [1].

The ongoing adaptation of mosquito populations to genetic pressures, such as the widespread use of insecticide-treated bed nets, indoor residual spraying [2] and agricultural insecticides [3,4], complicates efforts to control malaria transmission. This is particularly challenging for *Anopheles gambiae* sensu lato (s.l.), the primary vector of malaria in sub-Saharan Africa, which is highly heterogeneous and complex in its population structure. *An. gambiae* s.l. comprises seven morphologically indistinguishable species that vary in distribution, ecology, their role in malaria transmission. Among these, *An. gambiae* and *An. coluzzii* are the most prominent species in West Africa, including Côte d'Ivoire, where their distribution is highly variable across ecological zones and has been shown to influence insecticide resistance profiles and transmission dynamics [5–8].

Insecticide resistance is pervasive among *Anopheles* mosquitoes in sub-Saharan Africa [9,10]. Pyrethroid resistance, in particular, has become widespread due to the extensive use of long-lasting insecticidal nets (LLINs), a cornerstone of malaria

vector control. The massive scale-up of LLINs has led to a significant increase in pyrethroid resistance in mosquito populations in Côte d'Ivoire [5,11]. Additionally, resistance to other insecticide classes, including carbamates and organophosphates, has been documented in *An. gambiae* s.l. across various ecological settings [5,12–15], indicating a high intensity of insecticide resistance within this species complex [6,16]. Several studies conducted in Côte d'Ivoire have shown that resistance patterns vary significantly between *An. gambiae* complex species and across ecological zones, highlighting the importance of species-specific analysis in resistance monitoring and control planning [5,17,18]. While previous studies have documented insecticide resistance across various settings in Côte d'Ivoire, few have explicitly addressed how species diversity within the *An. gambiae* complex influences resistance profiles across ecological zones.

Environmental and ecological variations have been shown to influence mosquito species diversity [5,18–20] and their susceptibility to insecticides [3,4,21]. Understanding how genetic variation within vector populations affects the expression of insecticide resistance is needed for optimizing vector control strategies and developing more effective malaria control tools. In this study, we access the species diversity within the *An. gambiae* complex, as well as their distribution and levels of insecticide resistance across three distinct ecological zones in Côte d'Ivoire.

## Methods

### Study sites and design

A cross-sectional study was conducted in six sentinel sites established by the National Malaria Control Programme (NMCP) spanning three ecological zones across Côte d'Ivoire: Korhogo in the savannah (wooded savannah) zone; Man, Bouake and Sakassou in the pre-forest (transition) zone; Aboisso and Bassam in the forest (rainforest) zone (Fig 1). The map was generated using QGIS v2.14.19 with GPS coordinates of the collection points and OpenStreetMap (OSM) base layers, which are freely available under the Open Database License (ODbL). All vector layers, including vegetation and administrative boundaries, were digitized by the authors In each study site, we assessed the bionomics of malaria vectors, including species diversity, and determined insecticide resistance patterns. Specifically, we compared the distribution of *An. gambiae* s.l. species across ecological zones to capture variations in species diversity and insecticide resistance.

### Ethical consideration

The survey was reviewed and approved by the National Ethics Committee for Life Sciences and Health of the Ministry of Health, Public Hygiene, and Universal Health Coverage in Côte d'Ivoire (reference number 040–20/MSHP/CNESVS-kp). Community consent was first obtained from village leaders. Written informed consent was collected from heads of households, allowing CDC light traps to be installed on their premises, and from local volunteer supervisors, who worked with medical entomologists and technicians to oversee the deployment and operation of the traps. Study information sheets outlining the objectives and procedures were provided to all participants before consent was obtained. This national ethical approval, along with written informed consent from community members, served as the official authorization to conduct all mosquito collections at the study sites.

### Mosquito sampling

From July to October 2020, larval and adult mosquitoes were sampled across all six study sites. Larvae were collected from various breeding sites in both urban and rural areas using standard larval dippers. *An. gambiae* s.l. larvae and pupae collected were pooled by study site and reared to adulthood at the Institut Pierre Richet insectary in Bouaké. Laboratory conditions were maintained at a temperature of 27°C ± 2°C and relative humidity of 75% ± 20%. Adult mosquitoes were collected using CDC light traps, which were set for two to three nights per study site. On each night, traps were installed in five different households. In each household, one trap was placed indoor (near a bed) and one outdoor (on the veranda), totaling ten traps per night per site. Households were rotated on each collection night to ensure spatial representativeness

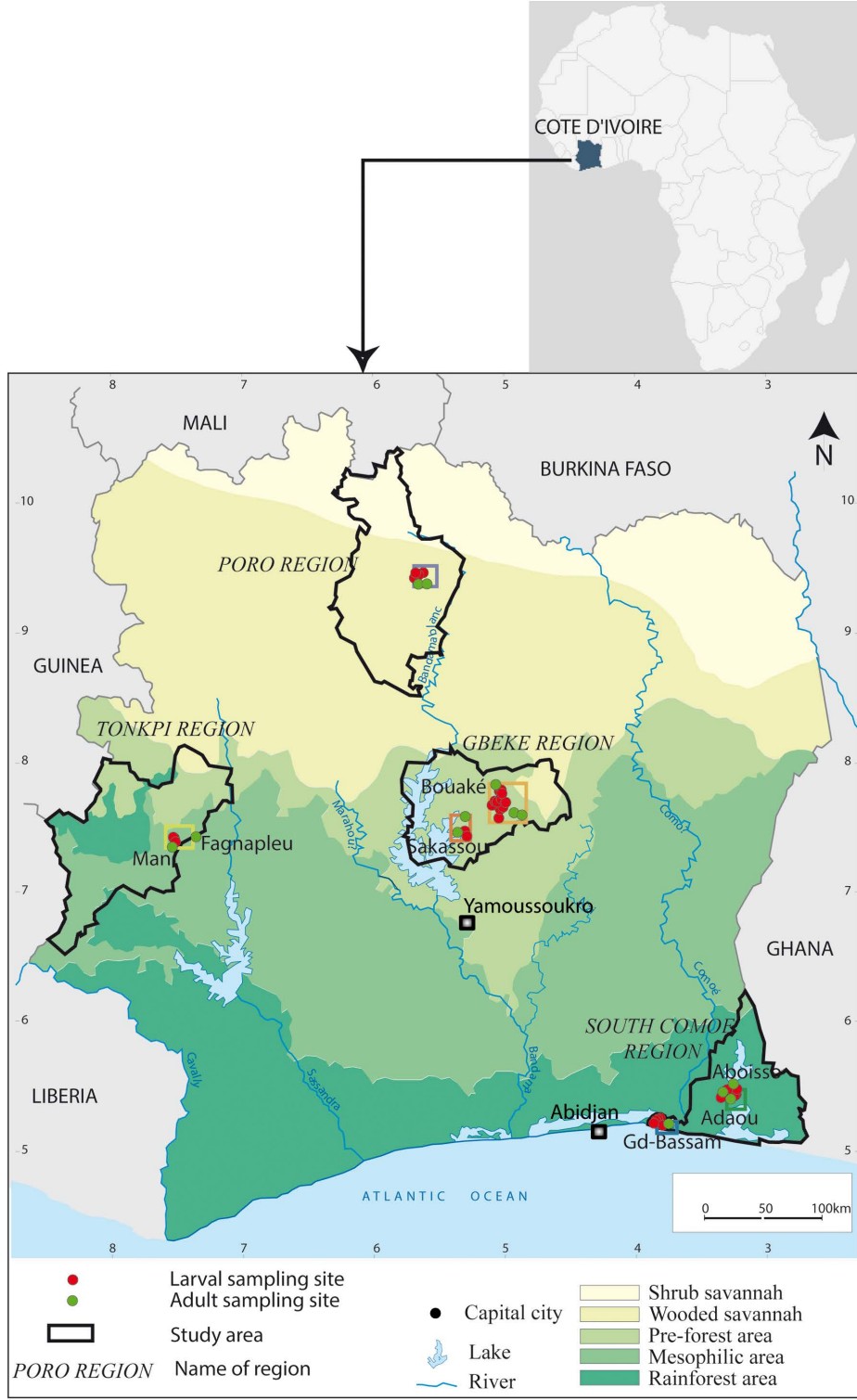

Source : OpenStreetMap and Wolie, 2020

**Fig 1. Map of Côte d'Ivoire showing the study sites.** Locations of the six sentinel sites showing larval and adult mosquito sampling points within three ecological zones: savannah (wooded savannah), pre-forest (transition), and forest (rainforest). The map was created with QGIS v2.14.19 using GPS coordinates and OpenStreetMap (OSM) base layers (Open Database License).

of mosquito sampling across the site. Traps were switched on at 18:00 in the evening and switched off at 08:00 in the morning the following day. Female mosquitoes were identified morphologically using species identification keys described by Gillies and Meillon [22]. *Anopheles* malaria vectors were counted and stored in silica gel for further molecular analysis.

## WHO bioassays for insecticide resistance

WHO insecticide susceptibility tests were conducted on 3–5 day old adult female *An. gambiae* s.l. that were reared from larval collections [23]. Insecticide-impregnated papers were obtained from Universiti Sains Malaysia (USM). First, mosquitoes were tested against papers impregnated with the diagnostic doses of the following insecticides: the two pyrethroids; deltamethrin (0.05%) and permethrin (0.75%), and the organophosphate pirimiphos-methyl (0.25%). Second, synergist tests were conducted by exposing mosquitoes to papers impregnated with 4% piperonyl butoxide (an inhibitor of oxidases) [24] for one hour before exposure to deltamethrin and permethrin at their diagnostic doses. Third, mosquitoes were tested against five and ten times the diagnostic doses of the insecticides: deltamethrin at 0.25% and 0.5%, permethrin at 3.75% and 7.5% and pirimiphos-methyl at 1.25% and 2.5%. Papers treated with silicon oil served as controls for pyrethroids and papers treated with olive oil served as controls for organophosphates.

Delayed mortality was recorded 24 hours after exposure to insecticides, and the insecticide resistance status of mosquitoes was determined according to WHO guidelines [23]. Following exposure to the diagnostic dose, mortality rates less than 90% indicated resistance, between 90 and 98% required further testing (under the same conditions) to confirm resistance status, and higher than 98% indicated susceptibility. Following exposure to five times the diagnostic dose, mortality rates between 90 and 98% were considered as low resistance. If mortality rates were less than 98%, mosquitoes were then exposed to ten times the diagnostic dose. If mortality rates at ten times the diagnostic dose were between 90% and 98%, the resistance intensity was classified as moderate. If mortality rates were below 90%, resistance intensity was considered high. For synergist assays, increased mortality after pre-exposure to PBO, compared to a diagnostic dose to the insecticide alone, indicated the overexpression of enzymes (such as P450s) in the test populations.

## Species identification and target site resistance mechanisms

A total of 504 *An. gambiae* s.l., females (42 per sentinel sites and per capture method), were randomly selected, for species identification and genotyping of *Kdr* L1014F and *Ace-1$^R$* G119S mutations. Genomic DNA was extracted from a pair of legs taken from *An. gambiae* s.l. mosquitoes using cetyl trimethyl ammonium bromide (CTAB) [25]. SINE PCR was used to identify members of the *An. gambiae* species complex [26]. The presence of the voltage-gated sodium channel mutation kdr L1014F within each *An. gambiae* s.l. population was also determined using a TaqMan protocol [27]. The presence of the acetylcholinesterase *Ace-1$^R$* G119S mutation was determined using a Taqman assay, as above [28].

## Statistical analysis

For each insecticide dose, Chi-squared tests were performed in R (v.4.0.3) to compare mortality rates between study zones. Results were presented in terms of changes in mosquito mortality. Values of $p \leq 0.05$ were considered statistically significant.

Allelic frequencies of resistance genes (*Kdr* L1014F and *Ace-1$^R$* G119S) were calculated as follows: $F(R) = (RS + 2(RR)) / (2(RS + RR + SS))$. These mutations comprise three genotypes expressing different allelic variants on the targeted loci, where the resistant (R) and susceptible (S) alleles are possible versions of these genes: RR indicates the resistant homozygous genotype; RS, the heterozygous genotype; and SS, the susceptible homozygous genotype. Fisher's exact tests were performed in MedCalc (https://www.medcalc.org/calc/comparison_of_proportions.php) to compare species and resistance allele distribution between study areas [29].

## Results

### Wild mosquito density and composition

In total, CDC light traps captured 1,982 mosquitoes in all study areas. Malaria vectors accounted for 39.2% of the total mosquito population, with 32.7% identified as *An. gambiae* s.l., 4% as *An. funestus*, and 2.5% as *An. nili.* Other culicidae were presents, in particular *Culex sp* (16.8%), *Aedes* sp (0.4%) and the most abundant *Mansonia sp* (42.5%) (Fig 2).

### WHO susceptibility assays

Across all areas, mortality rates of *An. gambiae* s.l. to deltamethrin and permethrin were below 10%, indicating severe resistance to pyrethroids (Fig 3). Increasing 5 or 10 times the diagnostic dose of deltamethrin also increased mortality rates of *An. gambiae* s.l. although values never exceeded 60%. Resistance intensity with permethrin was less, with 10 times permethrin diagnostic dose killing between 62–75%. Pre-exposure of mosquitoes to piperonyl butoxide (PBO) induced little additional mortality (less than 40%) indicating major resistance mechanisms other than P450. At a lesser extent, there was resistance to pirimiphos methyl; induced mortality to the diagnostic dose ranged 67–75%. Ten times the diagnostic dose pirimiphos methyl fully restored the susceptibility of this molecule (Fig 3).

### Species composition of the *An. gambiae* s.l. in three ecological zones

Out of 504 mosquitoes analyzed by qPCR, 486 were successfully identified, yielding a failure rate of less than 5%. Three hybrids of *An. gambiae*/ *An. coluzzii* were found among adults reared from larvae collected in the forest area (Aboisso). These were excluded from the analysis comparing species distribution. Mosquitoes collected in the savannah area, either at larval or adult stage were essentially *An. gambiae* (>97%), with only one mosquito identified as *An. coluzzii* (2.4%).

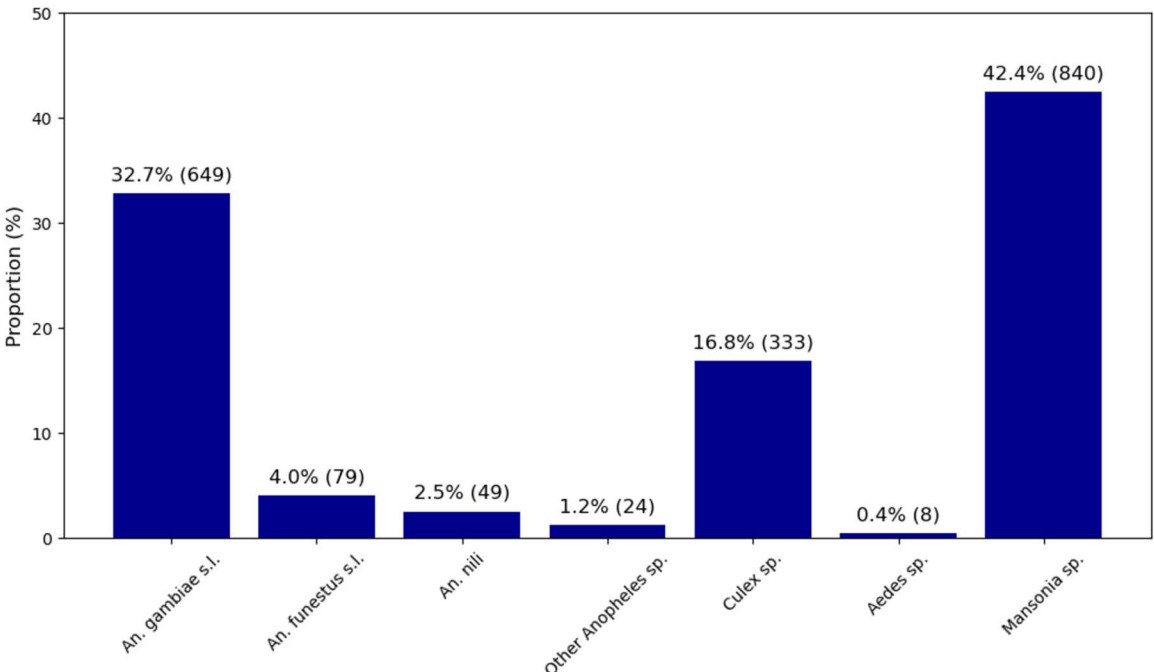

**Fig 2. Composition and density of adult mosquitoes captured using CDC-type light traps between July and October 2020.**

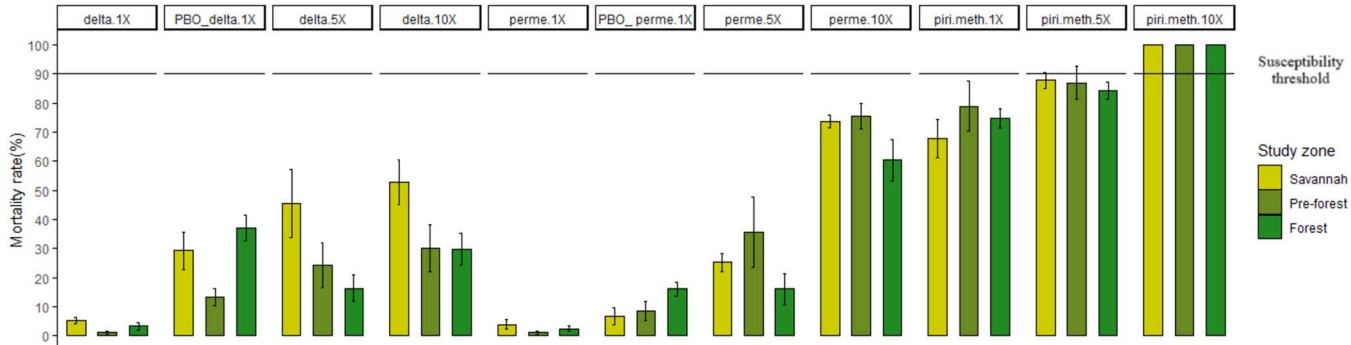

**Fig 3. Susceptibility of *An. gambiae* s.l. to insecticide doses alone or with synergist in the three ecological zones between July and October 2020.** Error bars indicate 95% confidence intervals. The dotted line at 90% mortality indicates WHO susceptibility threshold. delta = deltamethrin, perme = permethrin, pri.meth = pirimiphos methyl, PBO = piperonyl butoxide, nX = n times the diagnostic dose.

Reversely, in the pre-forest area, *An. coluzzii* was dominant (90–96%), but *An. coluzzii* and *An. gambiae* were both found in similar proportion in the forest areas (27–65% *An. gambiae* vs 34–72% *An. coluzzii*) (Table 1).

Species composition between larval and adult catches differed by study area. Composition between larvae and adults catches were similar for *An. gambiae* and *An. coluzzii* in savannah and pre-forest areas (P > 0.05) but varied significantly in the forest area. For example, there were more *An. gambiae (*65.8%) collected at adult stage than at larval (27.7%) ($\chi^2$ = 8.9; P = 0.003), whereas the situation was reversed with *An. coluzzii,* i.e., the proportion collected at larval stage (72.3%) was greater than that collected as adult (34%) (Table 1).

### Distribution of target site resistance mechanisms in the sibling species of *An. gambiae* s.l.

Overall, *Kdr* L1014F allele frequencies were higher in *An. gambiae* (82–92%) than in *An. coluzzii* (58.5–61.8%) across all study areas. The trend was similar for the *Ace-1^R* G119S gene between the two species (Table 2, S1 Table). Overall, there was trend for higher frequency of *Kdr* L1014F among adult *An. gambiae* and *An. coluzzii* emerged from larvae than in those caught directly as adult (Table 2). In the forested area where mixed populations (*An. gambiae + An. coluzzii*) were mostly found in sympatry from larval or adult source, the frequencies of *Kdr* L1014F were lower than in those from the savannah (*An. gambiae*) and pre-forest (mostly *An. coluzzii*) areas (Table 2, S1 Table).

The *Ace-1^R* G119S mutation was also detected in *An. gambiae* and *An. coluzzii*, albeit at lower frequencies. The overall frequencies of the resistant allele were significantly higher in *An. gambiae* (28.9–30.9%) than in *An. coluzzii* (13.2–17.6%) (P < 0.05) (Table 2, S1 Table).

**Table 1. Species composition of *An. gambiae* s.l. from larval and adult collections in three ecological areas between July and October 2020.**

| | *An. gambiae* | | *An. coluzzii* | |
|---|---|---|---|---|
| | % species (N) | | % species (N) | |
| | Larval catches | Adult catches | Larval catches | Adult catches |
| **Savannah** | 100(41)[a,1] | 97.6(41)[a,1] | 0(0)[a,1] | 2.4(1)[a,1] |
| **Pre-forest** | 9.8(12)[b,1] | 4.3(5)[b,1] | 90.2(111)[b,1] | 95.7(112)[b,1] |
| **Forest** | 27.7(33)[c,1] | 65.8(27)[c,2] | 72.3(86)[c,1] | 34.2(14)[c,2] |

N = Number of mosquitoes

For each species, superscript letters (a, b, c) indicate comparisons between ecological zones for the same life stage (larval or adult), while superscript numbers (¹, ²) indicate comparisons between larval and adult stages within the same ecological zone. Values sharing the same letter or number do not differ significantly (P > 0.05, 95% confidence of interval (CI)). Hybrids *An. coluzzii/ An. gambiae* were not included in the comparisons.

**Table 2. Variation in the allelic frequencies of Kdr L1014F and Ace-1$^R$ G119S mutations in An. gambiae s.l. in three ecological areas between July and October 2020.**

| Zones | An. gambiae | | | | An. coluzzii | | | |
|---|---|---|---|---|---|---|---|---|
| | Kdr L1014F Freq (N) | | Ace-1$^R$ G119S Freq (N) | | Kdr L1014F Freq (N) | | Ace-1$^R$ G119S Freq (N) | |
| | Larval catches | Adult caches | Larval catches | Adult caches | Larval catches | Adult caches | Larval catches | Adult caches |
| Savannah | 100.0 (39) [a,1] | 100.0 (41) [a,1] | 32. 9 (38) [a,1] | 26. 9 (39) [a,1] | – | - (1) | - (0) | - (1) |
| Pre-forest | 100.0 (12) [a,1] | 100. 0 (5) [a, 1] | 8. 3 (12) [b, 1] | 20. 0 (5) [a,1] | 83. 3 (111) [a,1] | 58. 6 (110) [a,2] | 15.7 (108) [a,1] | 16. 4 (110) [a,1] |
| Forest | 78.1 (32) [b,1] | 51. 8 (27) [b, 2] | 37. 1 (31) [a, 1] | 33. 3 (27) [a,1] | 63. 1 (84) [b,1] | 57. 1 (14) [a,1] | 10. 0 (85) [a,1] | 28. 6 (14) [a,2] |
| Total | 91.6 (83) | 82. 2 (73) | 30. 9 (81) | 28. 9 (71) | 61. 8(195) | 58. 5 (124) | 13. 2 (193) | 17. 6 (125) |

N. Number of mosquitoes, Freq: Frequency

For each mutation in each species, superscript letters indicate comparisons between ecological zones for the same life stage (larval or adult), while superscript numbers indicate comparisons between larval and adult stages within the same ecological zone. Values sharing the same superscript letters do not differ significantly. (P > 0.05; 95% confidence interval). An. coluzzii/ An. gambiae hybrids were not taken into account during the comparisons.

## Discussion

This study explored the species diversity and distribution of insecticide resistance, including high-intensity pyrethroid resistance, in natural *An. gambiae* s.l. populations across three distinct ecological areas in Côte d'Ivoire.

This study revealed the presence of both *An. gambiae* and *An. coluzzii* in sympatric populations across different ecological areas in Côte d'Ivoire, consistent with previous studies conducted in the region [6,13,30]. Interestingly, the detection of hybrid specimens, although rare and limited to the forest zone, the detection of hybrid specimens in Aboisso confirms previous findings reported in the same locality [18]. In that study, conducted from January 2018 to December 2020 using larval sampling, hybrids represented 4.17% of the *An. gambiae* s.l. population. In the current study, where the identified hybrids were also obtained from larval collections over a shorter period (July to October 2020), they accounted for only 0.62% of identified specimens. This lower proportion, despite the similar collection method, further highlights the rarity of hybrids in this locality. Although their reported to be fertile and can be sustained under laboratory conditions [31], hybrids remain relatively scarce in natural settings. This persistent rarity may reflect ecological and evolutionary constraints on hybrid survival, competitiveness, or reproduction in natural environments. Further research is needed to elucidate the mechanisms limiting their establishment and spread in the wild [32,33].

The high intensity of pyrethroid resistance observed in this study, particularly in areas mostly populated by *An. coluzzii*, highlights the key issue that insecticide resistance threatens malaria control efforts. The elevated resistance levels in *An. coluzzii* populations may be driven by intense selection pressure exerted by widespread insecticide use for agricultural purposes in these areas [34–36].

The partial restoration of susceptibility following pre-exposure to PBO suggests that additional mechanisms other than cytochrome P450 enzymes play significant role in the observed resistance pattern [5]. The presence of P450 enzymes was also detected in *An. coluzzii* from two transition areas (Bouaké and Sakassou) [5,6,16] and in a forested area (Tiassalé) [37,38].

Furthermore, The presence of higher *Kdr* L1014F allele frequencies in *An. gambiae* (82–92%) compared to *An. coluzzii* (58.5–61.8%) is consistent with observations from many other parts of Côte d'Ivoire [7,18,39] and elsewhere in Benin [40]. This current finding suggests that resistant specimens of *An. gambiae* are better adapted to pyrethroid insecticides pressure than *An. coluzzii*, in natural population.

The differences in species composition between larval and adult stages, particularly in the forest area, where *An. gambiae* was more prevalent at the adult stage while *An. coluzzii* dominated among the larval collection, suggest possible ecological partitioning or differential survival rates between the two species. This pattern may indicate that *An. gambiae* adults have a better adaptive condition in that forested environment. Similar life-stage shifts in species dominance have

been reported in other African settings and attributed to factors such as predation, larval habitat stability, and adult survival strategies [41,42]. The higher proportion of *An. gambiae* in adult collections could also point to a greater anthropophilic tendency in this species, which has implications for malaria transmission dynamics, particularly in forested areas where human-vector contact is frequent [43]. However, the observed differences may partly reflect the study's limited timeframe (July-October), which does not capture seasonal variation, and the unequal number of sites per ecological zone, especially in the savannah. Future studies should adopt year-round and more balanced sampling to better assess species dynamics.

The findings of this study underline the importance of tailoring malaria control strategies to specific ecological and species composition contexts in different areas. The high levels of pyrethroid resistance in *An. gambiae* s.l. populations, especially in areas where *An. coluzzii* predominates, pose significant challenges to the use of standard LLINs. Future research should focus on understanding the genetic and environmental factors that drive species diversity and resistance patterns in these populations.

## Conclusion

This study revealed the complex relationship between species diversity within the *Anopheles gambiae* complex and insecticide resistance across different ecological settings in Côte d'Ivoire. The coexistence of *An. gambiae* and *An. coluzzii*, along with the presence of rare hybrids, underscores the complexity of the population dynamics within *An. gambiae* s.l. The high levels of pyrethroid resistance, particularly in areas where *An. coluzzii* is dominant, highlight significant challenges to malaria control efforts.

The current findings indicate that species-specific differences and ecological factors contribute to varying resistance levels, emphasizing the need for tailored vector control strategies and resistance management. Future research should prioritize genomic analyses to better understand the evolutionary forces driving resistance and species diversity. This knowledge will be needed to develop more effective and sustainable malaria control strategies tailored to specific ecological and genetic contexts in different areas.

## Supporting information

**S1 Table. Variation in allelic frequencies of *Kdr* L1014F and *Ace-1*R G119S mutations within members of *An. gambiae* s.l. by study area.**
(DOCX)

## Acknowledgments

We sincerely appreciate the invaluable support provided by the technical staff at the Institut Pierre Richet, Bouake, Côte d'Ivoire, during the mosquito sampling surveys and laboratory analyses. Our special thanks go to Dr. Bamoro Coulibaly for their assistance in constructing the map of the study site in Côte d'Ivoire, which greatly enhanced the geographic context of this research. We also extend our gratitude to colleagues from various disciplines in Côte d'Ivoire, particularly those from the Unité de Recherche et de Pédagogie de Génétique, UFR Biosciences, Université Félix Houphouët-Boigny, Abidjan, for their significant contributions. Additionally, we thank the volunteer supervisors across the villages for their participation in this study.

## Author contributions

**Conceptualization:** Rosine Z. Wolie, Alphonsine A. Koffi, Joseph Chabi.

**Data curation:** Rosine Z. Wolie, Amal Dahounto.

**Formal analysis:** Rosine Z. Wolie.

**Funding acquisition:** Rosine Z. Wolie, Alphonsine A. Koffi, Joseph Chabi.

**Investigation:** Rosine Z. Wolie, Alphonsine A. Koffi, Kouakou P. M. Kouame, Florent H. A. Yapo, Soromane Camara, Innocent Z. Tia, Welbeck A. Oumbouke.

**Methodology:** Rosine Z. Wolie, Alphonsine A. Koffi, Joseph Chabi, Oulo N'Nan-Alla.

**Project administration:** Rosine Z. Wolie, Alphonsine A. Koffi, Joseph Chabi, Raphael N'Guessan.

**Software:** Rosine Z. Wolie.

**Supervision:** Rosine Z. Wolie, Alphonsine A. Koffi, Joseph Chabi, Raphael N'Guessan.

**Validation:** Rosine Z. Wolie.

**Writing – original draft:** Rosine Z. Wolie.

**Writing – review & editing:** Rosine Z. Wolie, Alphonsine A. Koffi, Joseph Chabi, Ludovic P. Ahoua Alou, Eleanore D. Sternberg, Kouakou P. M. Kouame, Florent H. A. Yapo, Soromane Camara, Innocent Z. Tia, Welbeck A. Oumbouke, Kallista Chan, Oulo N'Nan-Alla, Raphael N'Guessan, Amal Dahounto.

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
