## [Decision Letter · Decision Letter 0]

30 Jan 2025

Dear Dr. Wolie,

Thank you for submitting your manuscript to PLOS ONE. After careful consideration, we feel that it has merit but does not fully meet PLOS ONE’s publication criteria as it currently stands. Therefore, we invite you to submit a revised version of the manuscript that addresses the points raised during the review process.

We look forward to receiving your revised manuscript.

Kind regards,

Adekunle Akeem Bakare, Ph.D.

Academic Editor

PLOS ONE

Journal Requirements:

4. Please note that funding information should not appear in any section or other areas of your manuscript. We will only publish funding information present in the Funding Statement section of the online submission form. Please remove any funding-related text from the manuscript.

“This work was supported by a grant awarded to the Institut Pierre Richet / Institut National de Santé Publique, Côte d’Ivoire, from the Bill & Melinda Gates Foundation (OPP1210335) through the Pan African Mosquito Control Association (PAMCA). The grant aimed to strengthen malaria vector surveillance in Côte d’Ivoire, specifically addressing the genetic diversity and gene flow data gaps in insecticide-resistant Anopheles gambiae.”

6. In the online submission form, you indicated that “The data supporting the conclusions of this manuscript are included within the text and are available from the corresponding author upon reasonable request.”

7. We note that Figure 1 in your submission contain map/satellite images which may be copyrighted. All PLOS content is published under the Creative Commons Attribution License (CC BY 4.0), which means that the manuscript, images, and Supporting Information files will be freely available online, and any third party is permitted to access, download, copy, distribute, and use these materials in any way, even commercially, with proper attribution. For these reasons, we cannot publish previously copyrighted maps or satellite images created using proprietary data, such as Google software (Google Maps, Street View, and Earth). For more information, see our copyright guidelines: http://journals.plos.org/plosone/s/licenses-and-copyright.

Reviewers' comments:

Reviewer's Responses to Questions

**Comments to the Author**

1. Is the manuscript technically sound, and do the data support the conclusions?

Reviewer #1: Yes

Reviewer #2: Yes

2. Has the statistical analysis been performed appropriately and rigorously?

Reviewer #1: Yes

Reviewer #2: Yes

3. Have the authors made all data underlying the findings in their manuscript fully available?

Reviewer #1: Yes

Reviewer #2: Yes

4. Is the manuscript presented in an intelligible fashion and written in standard English?

Reviewer #1: Yes

Reviewer #2: Yes

Reviewer #1: The manuscript is well structured and written.

My only comments are that it might have been a more valuable study if further investigation into the specifics of metabolic resistance was investigated e.g. which P450s are associated with the resistance recorded rather than the more general finding associated with the use of PBO.

Also, data on the susceptibility to newer insecticides that are being introduced for vector control uses such as chlorfenapyr, clothianidin and broflanilide might have been an interesting addition.

Reviewer #2: This study is about mosquito diversity and the distribution of resistance mechanisms among malaria vectors in three ecological areas in Côte d’Ivoire.

The overall manuscript was very well written in easy English, with good results and important recommendations.

I only have minor reviews which if addressed could help the readers to better understand the study.

Minor reviews

67 Please rephrase this sentence to make clearer when PBO was conducted

69-70 ‘target site mechanisms (Kdr L1014F and Ace-1R G119S) determined using PCR’ is there a ‘were’ that is missing?

75 please delete ‘collected’

76-77 ‘in the savannah area (97-100% of collected mosquitoes)’ and in all the result part of the background please change the range to the mean of the proportion with the confidence interval (CI). For example ‘in the savannah area (97.9%, 95% CI [96.2-99.7]) or simply rephrase

96-97 please add some references about the seven species

99 Please write some key findings of these authors in the text

108 the introduction is well written however it lacks some related literature that had already shown by their results the importance of addressing insecticides resistances patterns according to species level in Côte d’Ivoire and other country of Africa. Please add some of them.

It is not clear what the authors mean by ‘adequately’, please be more precise

109-110 add some references from the country of study

100, 122, 167 italic for malaria vector name Anopheles, please check throughout the paper

Method/Result

In the method section, it is important to mention how long the CDC light trap were installed (one week, one month, two days per week for two months, …?). This allows the reader to know the frequency of collection that allowed the authors to have the number of mosquitoes mentioned in the part of the results and help in terms of reproducibility of the study

129-132: ‘Adult mosquitoes were collected using CDC light traps, which were set for two to three nights per study site’. Does it mean that for each of the six sites chosen for this study just one CDC light trap was installed? Were they installed simultaneously in the sites or in different period? When talking about the three different locations (132) the authors only mentioned ‘near a bed and outside on the veranda’ is there any other location missing? Does it mean that there were three CDC light traps per house? This part of the method needs more details

162: the authors stated that 504 mosquitoes were chosen randomly to conduct species ID. The selection was random between each ecological region/ each method of capture? How did the authors proceed to avoid any bias? Please be more explicit

173 the authors mentioned odds ratios however throughout the manuscript there was no mention of OR. Please have a look

179 please add a link for the software

213-214 repetition of ‘both’ please delete one of them

Discussion

241-243 It was not the first time that hybrids were identified in Aboisso please check this reference https://doi.org/10.1371/journal.pone.0297604

243-247 you can review this part of the discussion considering the number of hybrids they found in this study https://doi.org/10.1371/journal.pone.0297604 or any other study that you can find from Aboisso, considering the date of collection and also the method of capture

259 Please add other references as you mentioned many other parts of Côte d’Ivoire

262-265. Anything about the low number of mosquitoes collected in the savannah region compared to the others? Don’t the authors think that the number of localities sampled by ecological regions could have influenced difference in terms of number of mosquitoes collected and the composition of mosquito species? Please address this either in the main discussion or in the limits of the study

I would like also the authors to address the limitations of the study regarding the period of collection July-October, because maybe the results would have been different when talking about species distribution in other periods of the year

487 Table 1. please add the stage of mosquitoes (larval and adult) in the title to make it more explicit

489 In the legend of the table 1 please be clearer about what type of significance the letters and numbers shown. For example, for each species in each column is the same letter shows that there is not a significant difference between method of capture? What about the numbers?

500 `Table’ instead of ‘tableau’

501 there is a ‘t’ missing in adult ‘catches’ in the table 2

503 Again as for the table 1, in the legend ‘For each mutation in each species, values in the same column sharing the same superscript letters do not differ significantly’ it is not clear what are the variables that the authors were addressing. Is it a significant difference between method of capture? Between ecological areas? Please address each parameter (letter/number) and add this information to the legend

Figure 2. instead of proportion (%) you can write proportion% (number)

Title of Figure 3. repetition of ‘synergist’

Legend of Figure 3. Please address the abbreviations as they are written in the figure. Repetition of what the errors bars indicate

**Do you want your identity to be public for this peer review?** For information about this choice, including consent withdrawal, please see our Privacy Policy

Reviewer #1: No

Reviewer #2: No

---

## [Author Response · Author response to Decision Letter 1]

29 Aug 2025

Response to reviewers and journal requirements

Responses to journal editorial requirements

Responses 1: We corrected the formatting of all section titles, updated the corresponding author asterisk, edited the inline equation, and revised the table titles, column headers, and row labels to comply with PLOS ONE guidelines.

2. In your Methods section, please provide additional information regarding the permits you obtained for the work. Please ensure you have included the full name of the authority that approved the field site access and, if no permits were required, a brief statement explaining why

Responses 2: We clarified in the Methods section that national ethical approval, together with written informed consent from community members, served as the official authorization to conduct mosquito collections at all study sites (Lines 206-208.).

Responses 3: We have removed the funding information from the main manuscript, as requested. It is now included only in the Funding Statement section of the online submission form (Lines 346).

4. Please note that funding information should not appear in any section or other areas of your manuscript. We will only publish funding information present in the Funding Statement section of the online submission form. Please remove any funding-related text from the manuscript.

Responses 4: This point duplicates the previous request. As stated above, we have removed the funding information from the main manuscript.

“This work was supported by a grant awarded to the Institut Pierre Richet / Institut National de Santé Publique, Côte d’Ivoire, from the Bill & Melinda Gates Foundation (OPP1210335) through the Pan African Mosquito Control Association (PAMCA). The grant aimed to strengthen malaria vector surveillance in Côte d’Ivoire, specifically addressing the genetic diversity and gene flow data gaps in insecticide-resistant Anopheles gambiae.”

Responses 5: The funders had no role in study design, data collection and analysis, decision to publish, or preparation of the manuscript. We will include this statement in the cover letter, as requested.

6. In the online submission form, you indicated that “The data supporting the conclusions of this manuscript are included within the text and are available from the corresponding author upon reasonable request.”

Responses 6: We have revised the Availability of data and materials section in the manuscript to align with PLOS ONE data sharing policy. It now states that the data are included in the text and that the underlying datasets will be deposited in a public repository and made fully accessible upon acceptance.

7. We note that Figure 1 in your submission contain map/satellite images which may be copyrighted. All PLOS content is published under the Creative Commons Attribution License (CC BY 4.0), which means that the manuscript, images, and Supporting Information files will be freely available online, and any third party is permitted to access, download, copy, distribute, and use these materials in any way, even commercially, with proper attribution. For these reasons, we cannot publish previously copyrighted maps or satellite images created using proprietary data, such as Google software (Google Maps, Street View, and Earth). For more information, see our copyright guidelines: http://journals.plos.org/plosone/s/licenses-and-copyright.

Responses 7: Thank you for this important clarification. The map was generated using QGIS v2.14.19 with GPS coordinates of the mosquito collection points. The base layers were obtained from the Centre de Cartographie et de Télédétection (CCT) of the Bureau National d’Études Techniques et de Développement (BNETD) in Côte d’Ivoire. All vector layers, including vegetation types and administrative boundaries, were digitized by the authors using publicly available sources. No copyrighted or proprietary maps (e.g., Google Maps) were used. This information has been added to the revised Methods section and the revised figure has been included in the resubmission (Lines 128-131).

Responses 8: We have added a Supporting information section at the end of the manuscript, including full captions for each supplementary figure, table, and document, in accordance with PLOS ONE guidelines (Lines 545-558).

Responses 9: We have reviewed the reference list and confirm that it is complete and up to date. No retracted articles are cited in the manuscript.

Responses to Reviewers’ comments:

Comments to the Author

1. Is the manuscript technically sound, and do the data support the conclusions?

Reviewer #1: Yes

Reviewer #2: Yes

2. Has the statistical analysis been performed appropriately and rigorously?

Reviewer #1: Yes

Reviewer #2: Yes

3. Have the authors made all data underlying the findings in their manuscript fully available?

Reviewer #1: Yes

Reviewer #2: Yes

4. Is the manuscript presented in an intelligible fashion and written in standard English?

Reviewer #1: Yes

Reviewer #2: Yes

5. Review Comments to the Author

Reviewer #1: The manuscript is well structured and written.

My only comments are that it might have been a more valuable study if further investigation into the specifics of metabolic resistance was investigated e.g. which P450s are associated with the resistance recorded rather than the more general finding associated with the use of PBO.

Also, data on the susceptibility to newer insecticides that are being introduced for vector control uses such as chlorfenapyr, clothianidin and broflanilide might have been an interesting addition.

Response to reviewer 1:

We appreciate the reviewer’s positive feedback and thoughtful suggestions. While our study focused on phenotypic resistance and target-site mutations, we acknowledge the importance of investigating specific metabolic resistance mechanisms, such as P450 gene expression. This limitation is now noted in the Discussion as a key area for future research.

Regarding newer insecticides like chlorfenapyr and clothianidin and broflanilide our study focused on insecticides for which resistance markers were well-characterized at the time, to examine how phenotypic resistance and known genetic markers are distributed across species and ecological zones. These markers are increasingly used to inform insecticide deployment and resistance management strategies. We also recognize the importance of evaluating susceptibility to these newer compounds, and this has been highlighted as a priority in our future research directions.

Response to reviewer 2:

Reviewer #2: This study is about mosquito diversity and the distribution of resistance mechanisms among malaria vectors in three ecological areas in Côte d’Ivoire.

The overall manuscript was very well written in easy English, with good results and important recommendations.

I only have minor reviews which if addressed could help the readers to better understand the study.

Minor reviews:

67 Please rephrase this sentence to make clearer when PBO was conducted

Response 1:

We thank the reviewer for this helpful observation. The sentence has been revised in the Abstract section and can be found at lines 66- 68 of the revised manuscript.

69-70 ‘target site mechanisms (Kdr L1014F and Ace-1R G119S) determined using PCR’ is there a ‘were’ that is missing?

Response 2:

We appreciate the reviewer’s attention to clarity. The sentence has been revised, and the updated version appears in the Abstract at line 71.

75 please delete ‘collected’

Response 3:

We thank the reviewer for this suggestion. The word “collected” has been removed, as recommended in the line 76.

76-77 ‘in the savannah area (97-100% of collected mosquitoes)’ and in all the result part of the background please change the range to the mean of the proportion with the confidence interval (CI). For example ‘in the savannah area (97.9%, 95% CI [96.2-99.7]) or simply rephrase

Response 4:

We have replaced the percentage ranges with mean values and 95% confidence intervals in the Abstract and can be found at line 77-83.

96-97 please add some references about the seven species

Response 5:

Thank you to the reviewer. A supporting reference has been added at line 102 and in the References section (lines 380-381).

99 Please write some key findings of these authors in the text

Response 6:

Thank you. Key findings from the cited studies have been added to contextualize species variation and distribution at lines 99-102.

108 the introduction is well written however it lacks some related literature that had already shown by their results the importance of addressing insecticides resistances patterns according to species level in Côte d’Ivoire and other country of Africa. Please add some of them. It is not clear what the authors mean by ‘adequately’, please be more precise

Response 7:

We thank the reviewer for this valuable suggestion. Relevant literature highlighting the importance of assessing insecticide resistance patterns at the species level in Côte d’Ivoire has now been added at lines 109-112. We have also clarified the intended meaning of “adequately” to better reflect the novelty of our study at lines 112-115.

109-110 add some references from the country of study

Response 8:

We thank the reviewer for this helpful suggestion. References from studies conducted in Côte d’Ivoire have now been added to strengthen this statement (line 117)

100, 122, 167 italic for malaria vector name Anopheles, please check throughout the paper

Response 9:

Thank you for the observation. All malaria vector names, including Anopheles species have been i

---

## [Editor Report · Decision Letter 1]

2 Sep 2025

Insecticide resistance and species diversity in Anopheles gambiae s.l. in Côte d’Ivoire

PONE-D-24-56516R1

Dear Dr. Wolie,

We’re pleased to inform you that your manuscript has been judged scientifically suitable for publication and will be formally accepted for publication once it meets all outstanding technical requirements.

Kind regards,

Adekunle Akeem Bakare, Ph.D.

Academic Editor

PLOS ONE
---

## [Editor Report · Acceptance letter]

PONE-D-24-56516R1

PLOS ONE

Dear Dr. Wolie,

I'm pleased to inform you that your manuscript has been deemed suitable for publication in PLOS ONE. Congratulations! Your manuscript is now being handed over to our production team.

Kind regards,

on behalf of

Professor Adekunle Akeem Bakare

Academic Editor

PLOS ONE